# REACHABILITY TRACES FOR CURRICULUM DESIGN IN REINFORCEMENT LEARNING

## ABSTRACT

The objective in goal-based reinforcement learning is to learn a policy to reach a particular goal state within the environment. However, the underlying reward function may be too sparse for the agent to efficiently learn useful behaviors. Recent studies have demonstrated that reward sparsity can be overcome by instead learning a curriculum of simpler subtasks. In this work, we design an agent's curriculum by focusing on the aspect of goal reachability, and introduce the idea of a reachability trace, which is used as a basis to determine a sequence of intermediate subgoals to guide the agent towards its primary goal. We discuss several properties of the trace function, and in addition, validate our proposed approach empirically in a range of environments, while comparing its performance against appropriate baselines.

## 1 INTRODUCTION

Reinforcement learning (RL) (Sutton & Barto, 1998) has been successfully used to train agents in domains such as robotics (Zhu et al., 2019), Atari games (Mnih et al., 2015) and other complex games (Silver et al., 2018). The universal principle underpinning these applications is the maximization of the long term expected rewards that the agent accumulates as it interacts with its environment. Following each interaction, the agent receives a numerical reward, which is (generally) directly or indirectly specified by a human user. The design of such reward functions is critical, as it fundamentally affects both the rate of learning (Matignon et al., 2006), as well as the type of behaviors learned.

Specifically, a sparse reward function makes learning challenging, as it deprives the agent of the necessary feedback required to improve its behavior. This has been a widely studied problem in RL, and although several potential solutions (Ng et al., 1999; Vecerik et al., 2017; Narvekar et al., 2020) have been proposed, it remains an active area of research.

Apart from sparsity, the reward function could also suffer from improper specifications, such as inappropriately chosen reward values for certain states or actions, which could distract the agent from its intended task. The consequences of such mis-specifications have been recorded in works such as Burda et al. (2018), where the RL agent, rewarded for curious behaviors, was shown to become distracted from its original task due to local sources of entropy (A TV with randomly changing channels). Similar effects have also been observed in Clark & Amodei (2016), where the agent, tasked with safely completing a boat race circuit, unexpectedly exploited the improperly designed reward function, and learned undesirable behaviors.

In this work, we propose the idea of *reachability traces*, which is based solely on the reachability of the goal state, and is independent of other aspects of the reward function. Similar to the idea of eligibility traces (Singh & Sutton, 1996) in classical RL, reachability traces model the temporal closeness (i.e., number of steps to the goal) of states/state-action pairs leading to the goal by assigning diminishing reachability values (traces) to these states, looking backwards from the goal state. These reachability values are approximated through a reachability trace function, which is realized through a simple feedforward neural network, updated online during learning. We also show that alternatively, reachability traces could be learned using an MDP (Markov Decision Process) framework (Puterman, 2014), guaranteeing its convergence in tabular environments. Once learned, the reachability trace provides an indication of the temporal closeness to the goal state, which is used to autonomously determine a sequence of achievable subgoals, which are subsequently learned. The sequence of subgoals are chosen in increasing order of their reachability trace values, which ensures that subgoals appearing later in the sequence are temporally closer to the goal state, and thus have a higher chance

of reaching the goal. The corresponding subpolicies, once learned, are used to provide the agent with action advice (Fachantidis et al., 2019), thereby guiding it towards the goal region. The use of action advice (accompanied by a non-zero probability of random exploration) to guide our off-policy agent implies the preservation of its convergence properties in tabular environments. We demonstrate our approach in sparse (discrete as well as continuous) goal based RL tasks and compare its performance against several other baselines. We also discuss and empirically demonstrate other use cases of reachability traces, such as in environments with poorly designed reward functions. In summary, the main contributions of this work are:

- The idea of reachability traces to model the temporal closeness to the goal.
- A framework for automatically discovering and learning reachable subgoals.
- An empirical comparison of our proposed approach to other baselines in a variety of reward sparse environmnents.

## 2    REACHABILITY TRACES

In goal based tasks, the aim of the agent is to learn a policy to reach a predetermined goal state. Depending on the reward function in question, the probability of reaching such a goal state may or may not be correlated with the (action-) value function of the agent. This is because the value function, which is solely designed to maximize the expected return, does not explicitly depend on goal state visits. For example, value function based learning would need to account for non-goal rewards, which (if improperly specified) could distract the agent from its intended task (Burda et al., 2018; Clark & Amodei, 2016). Particularly in environments where the goal rewards are sparse, we posit that it is more beneficial to exploit the rare trajectories that lead to the goal state, by learning about their goal-reaching properties, and subsequently using this knowledge to guide the agent's exploration. We posit that this goal-reaching property be characterized through reachability traces, which is solely based on the temporal distance to the goal state under the agent's behavior policy.

In classical RL, a concept that models the idea of temporal closeness is eligibility traces (Singh & Sutton, 1996). Although the primary motivation behind eligibility traces was to address the credit assignment problem, it was designed in a way that states/state-action pairs that were temporally more closely related to an event (a state or action), were assigned higher trace values compared to those that were temporally further away. Here, we use a similar idea to build a reachability trace function using historically successful trajectories. Unlike eligibility traces, reachability traces are learned as a neural network and are concerned solely with reaching the goal state, and modeling the temporal distance to the goal, which in this work, is used to evaluate the suitability of potential subgoals.

Assuming a behavior policy $\pi$, we consider a successful trajectory $T_S = \{s_t...s_{t+k}, ...s_{t+N}\}$ which terminates at $s_{t+N}$ (goal state). For each $s_{t+i} \in T_S$, we assign a reachability trace label $e_\pi(s_{t+i})$ as:

$$e_\pi(s_{t+i}) = \lambda^{N-i} e_0 \tag{1}$$

where $e_0$ (set to 1) is the highest possible trace label and $\lambda$ ($0 < \lambda < 1$) is the trace decay parameter. It is to be noted that traces are updated looking backwards from the terminal/goal state as per Equation 1. Trace labels are assigned to ensure that states temporally closer to the goal state are associated with higher trace values, and those further away are associated with lower values. States in unsuccessful trajectories are assigned a trace label of $0$. These assigned trace labels are then be used to learn a reachability trace function $\phi_\pi(s) : s \rightarrow e_\pi(s)$, which maps states (or state-action pairs) to their corresponding trace labels. Given enough successful trajectories, the learned reachability trace $\phi(s)$ would converge to the expected trace value:

$$\phi(s_t) = \mathbb{E}_\pi[\lambda^N e_0] \tag{2}$$

Our assumption is that prior to learning the trace function, our agent encounters at least one successful trajectory. We do not require that this trajectory be optimal; only that it enounters the goal state $G$ and terminates. As we obtain trace labels for multiple states upon visiting the goal, even a single successful trajectory could help build a rough estimate of the trace function. Our approach is also suitable to be used in the availability of (suboptimal) demonstrations, as described in Appendix D.

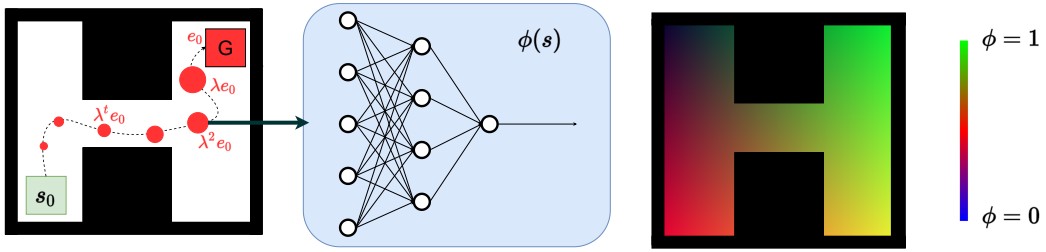

Figure 1: Illustration of a reachability trace network that maps states on the successful trajectory (left) of a hypothetical navigation environment to their trace labels.

---

**Algorithm 1** Learning a reachability trace function

1: **Input:**
2: Trace decay parameter $\lambda$, stored trajectory length $L$, trace function $\phi(s, \theta)$, goal state $G$
3: Initialize state $s = s_0$, successful trace buffer $D_{\phi_S}$ and unsuccessful trace buffer $D_{\phi_U}$
4: **Output:** Learned trace function $\phi$
5: **while** True **do**
6:     Interact with environment: take action $a$, observe $r, s'$
7:     Store latest $L$ states
8:     **if** Successful episode (G is reached) **then**
9:         **for** $i$ from 1 to $L$ **do**
10:             $e_i = \lambda^{L-i} e_0$
11:         **end for**
12:         Move the trace labels $e$ and corresponding states into the buffer $D_{\phi_S}$
13:     **else**
14:         **for** $i$ from 1 to $L$ **do**
15:             $e_i = 0$
16:         **end for**
17:         Move the trace labels $e$ and corresponding states into the buffer $D_{\phi_U}$
18:     **end if**
19:     Sample set of state and trace labels $(s_{samp}, e_{samp})$, obtaining each sample from either $D_{\phi_S}$ or $D_{\phi_U}$ with equal probability
20:     Minimize $(e_{samp} - \phi(s_{samp}, \theta))^2$ with respect to trace network parameters $\theta$
21:     $s \leftarrow s'$
22: **end while**

---

Since the trace function is indicative of closeness to the goal state $G$, it is used as a basis for generating subsequent subgoals that are likely to be closer to $G$. In this work, the reachability trace is realized through a simple feedforward neural network, trained on data of states and corresponding trace labels generated from successful and unsuccessful trajectories, which are stored in separate buffers. The steps to be followed to learn a trace function have been described in Algorithm 1. Here, successful and unsuccessful trajectories are sampled with equal probability, which causes the learned trace to be a biased estimate of that mentioned in Equation 2. Despite this bias, we find that the learned reachability trace serves as an approximate measure of temporal proximity, and is still a useful basis for selecting subgoals. In addition, as our approach does not require generated suboals to lie on the optimal path, the use of this biased estimate is justified. Although reachability traces have been described only in terms of states (for the sake of simplicity), similar to the case in eligibility traces, it can also be extended to state-action pairs ($\phi_\pi(s, a)$).

## 3 SUBGOAL GENERATION AND CURRICULUM BUILDING

Once an appropriate trace function $\phi$ has been learned, we use it to identify and learn potential subgoals of increasing difficulty. The intuition is that when goal rewards are sparse, a high value

of $\phi(s)$ or $\phi(s,a)$ indicates that the state is temporally close to the goal state, and thus may be less reachable (from the starting state), while a low $\phi$ value may indicate that the state is further away from the goal state (and thus temporally closer to the starting state), and thus may consitute an easier subgoal to be learned. In this work, we treat this temporal proximity as a proxy for subgoal difficulty, and use it as a basis for building a task curriculum (Portelas et al., 2020; Narvekar et al., 2020) in sparse reward problems. The idea is to first learn simple subgoals (low $\phi$) and progressively move to more difficult ones till the main task is solved.

We consider a sparse-reward MDP (main task) $\mathcal{M} = \{\mathcal{S}, \mathcal{A}, \mathcal{R}, \mathcal{T}, G, s_0\}$, with state space $\mathcal{S}$, action space $\mathcal{A}$, reward function $\mathcal{R}$, and transition function $\mathcal{T}$. $s_0$ and $G$ represent the start and terminal goal states respecitvely. Under the standard learning scenario, learning an optimal policy $\pi^*$ to solve this sparse reward MDP may become prohibitively sample inefficient. However, if we have learned a trace function $\phi$, $\mathcal{M}$ can be simplified into a sequence of subtasks $\mathcal{M} = [M_0...M_i...M_N]$ such that each $M_i \in \mathcal{M}$ is a subtask MDP $M_i = \{\mathcal{S}, \mathcal{A}, R_i, T_i, g_i, s_i\}$ sharing the same state-action space, with its own reward function $R_i$, start state $s_i$ and terminal goal state $g_i$. The starting state for the subsequent subtask is set as the goal state of the previous one; that is, $s_i = g_{i-1}$. The transition function $T_i$ is assumed to be identical to $\mathcal{T}$, except that as per $T_i$, $g_i$ is terminal, whereas $\mathcal{T}$ assumes termination at $G$. The subtask sequence is such that for any two subtasks $M_i$, $M_j$, where $j > i$, $\phi(g_j) > t_\phi\phi(g_i), t_\phi (\geq 1)$ is a subgoal trace scaling factor. Choosing large values of $t_\phi$ would result in subgoals spaced temporally farther, while smaller values of $t_\phi$ could be expected to return subgoals that are temporally close to one another. Guidelines for selecting $t_\phi$ are described in Appendix B. The reward $r_i(s)$ corresponding to the reward function $R_i$ for each subtask is binary, such that:

$$r_i(s) = \begin{cases} 1 & s = g_i \\ 0 & otherwise \end{cases} \tag{3}$$

In order to identify subgoals online, the agent is allowed to interact with the environment in an episodic manner, till it encounters a state $s_{max}$ corresponding to the highest $\phi$ value experienced during the episode, which also exceeds the trace value of the latest subgoal state $g_i$. That is, if $\phi(s_{max}) > t_\phi\phi(g_i)$, $s_{max}$ is considered to be the new subgoal ($g_{i+1}$), and accordingly, subtask $M_{i+1}$ is generated with high rewards corresponding to state $g_{i+1}$, and 0 rewards for all other states. For each such subtask, a corresponding subpolicy is learned, following which the same process is repeated to generate the next subtask, and so on till the goal state $G$ is reached. The fact that only previously visited states are chosen as subgoals ensures that the chosen subgoals are indeed reachable. At this point, the main task $\mathcal{M}$ would be decomposed into several subtasks consisting of subgoals of progressively increasing difficulty (Figure 6, Appendix B), whose subpolicies are learned in sequence. Although these subpolicies are used to guide the agent towards the goal (described in Section 4), they can also potentially be reused (Fernández & Veloso, 2006) to aid transfer learning (Appendix E). The processes involved for subgoal generation and curriculum building are described in Algorithm 2.

## 4 Curriculum Guided Learning

Once a curriculum $C$ of subpolicies have been learned as described in Algorithm 2, the learned subpolicies are simply used in sequence to guide the exploration of the agent till the goal state $G$ is reached. It is to be noted that although the approach described in Algorithm 2 generates subgoals of increasing difficulty/trace values, it does not guarantee that they lie on the optimal path to the goal state $G$. However, in this work, as the subpolicies only serve to guide the agent via action advice, it is guaranteed to eventually converge (in the tabular case) to the optimal policy.

To learn the main policy, the corresponding $Q-$function $Q(s,a)$ is updated online with off policy updates (DQN or DDPG for instance), using guided actions from the learned subpolicies $Q_{sub}$. The process of action advising involves extracting greedy actions from the relevant subpolicy $Q_{sub}$, and executing it with a probability $\epsilon$ (while maintaining a small non-zero probability $\delta > 0$ of taking random actions), and otherwise acting greedily with respect to $Q$. The requirement of $\delta > 0$ is enforced to ensure sufficient exploration in the case of highly suboptimal subgoals. Once the subgoal of the corresponding subpolicy is reached, the agent receives subsequent advice from the next subpolicy, and so on till the main goal $G$ is reached, with $Q(s,a)$ being updated with off-policy updates at every step. The overall curriculum-guided learning is described in Algorithm 3.

---

**Algorithm 2** Subgoal generation and curriculum building

---

1: **Input:**
2: discount factor $\gamma$, goal state $G$, subgoal trace scaling factor $t_\phi$, No. of episodes $N_e$
3: state $s = s_0$, Initialize agent $Q-$ function $Q(s,a)$, $Full\_curriculum\_found \leftarrow 0$, $\phi_{prev} = 0$,
   Initialize curriculum $C$, initialize list of all subgoal states $g_{all} = [\ ]$
4: **Output:** Curriculum $C$, subgoal states $g_{all}$
5: **while** $Full\_curriculum\_found == 0$ **do**
6:     Interact with environment: take action $a$, observe $r, s'$
7:     Update $Q(s,a)$
8:     **if** $G$ is reached **then**
9:         Update trace function $\phi(s, \theta)$ using Algorithm 1
10:    **end if**
11:    Initialize subpolicy $Q-$function $Q_{sub}$; $subgoal\_found = 0$
12:    **for** $i$ ranging from 1 to $N_e$ **do**
13:        **if** $subgoal\_found == 0$ **then**
14:            Execute episode $i$
15:            Find state $s_{max}$ from episode $i$ corr. to maximum $\phi$ value
16:            **if** $\phi(s_{max}) > t_\phi \phi_{prev}$ **then**
17:                $s \leftarrow g; g \leftarrow s_{max}; subgoal\_found = 1; g_{all} \leftarrow g_{all} \cup g$
18:                $\phi_{prev} = \phi(s_{max})$
19:                **if** $g == G$ **then**
20:                    $Full\_curriculum\_found = 1$
21:                **end if**
22:            **end if**
23:        **else**
24:            Update $Q_{sub}$ during episode $i$
25:        **end if**
26:    **end for**
27:    $C \leftarrow C \cup Q_{sub}$
28: **end while**

---

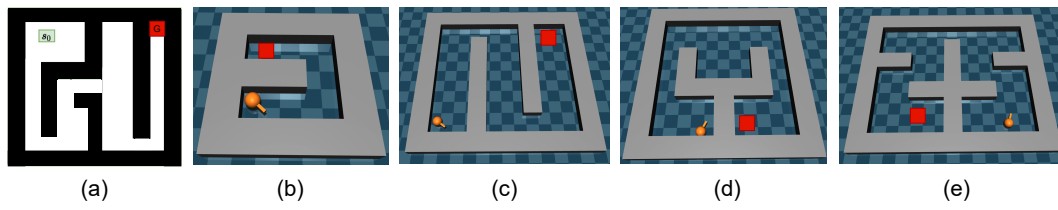

Figure 2: The (a) Gridworld environment (b) U-shaped, (c) S-shaped, (d) $\omega$-shaped and (e) $\Pi$-shaped maze environments used for evaluation, where the red marker denotes the goal position.

## 5 EXPERIMENTS

We evaluate our proposed approach first on a simple tabular environment shown in Figure 2 (a), followed by a number of continuous state and continuous action environments shown in Figure 2 (b)-(e), which were previously also used in (Chane-Sane et al., 2021).

The environment in Figure 2 (a) consists of a $Q-$learning (Watkins, 1989) agent starting at the state $s_0$, with a goal state $G$. The agent can take actions to move in the four cardinal directions (up, down, left and right). It receives a reward of $0$ for each transition, except those that lead to the terminal goal state, for which it receives a reward of $+1$. When it encounters an obstacle, the agent's state remains unchanged. Episodes terminate after a fixed episode horizon or upon reaching the goal state. The agent interacts with the world using an $\epsilon$-greedy strategy, with a linearly decaying $\epsilon$, and the agent's state is reset after each episode. Other associated hyperparameters are specified in Appendix H. In this sparse reward setting, our approach first interacts with the environment till the goal state is visited, following which it learns (Algorithm 1) a trace function (depicted in Figure 3 (center). Using

---

**Algorithm 3** Curriculum-based action advising

1: **Input:**
2: Learned curriculum $C$, maximum steps $N_{max}$, agent's $Q-$ function $Q(s, a)$, goal state $G$, set of subgoals $g_{all}$, goal index $k = 0$
3: state $s = s_0$, $step \leftarrow 0$, $Q_{sub} = C[0]$, $g = g_{all}[k]$
4: **Output:** Optimal $Q-$function $Q^*$
5: **while** $step \leq N_{max}$ **do**
6:    **if** $g$ not found **then**
7:       Get action $a_{sub}$ from $Q_{sub}$
8:       **if** $\epsilon >$rand() **then**
9:          Execute $a_{sub}$ (random exploration with a small probability $\delta$)
10:       **else**
11:          Greedy action w.r.t. $Q$
12:       **end if**
13:       Observe $r, s'$ resulting from the executed action
14:       $step = step + 1$
15:    **else**
16:       $k = k + 1$
17:       $Q_{sub} = C[k], g = g_{all}[k]$
18:    **end if**
19:    Update $Q$ using an off-policy update (Eg: $Q-$learning)
20:    **if** G is reached **then**
21:       $s \leftarrow s_0$
22:    **else**
23:       $s \leftarrow s'$
24:    **end if**
25: **end while**

---

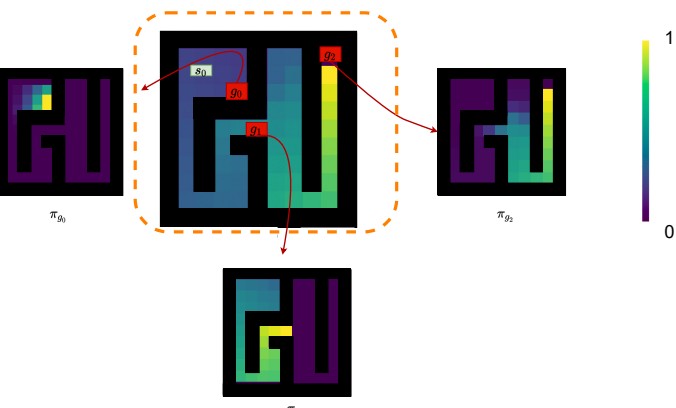

Figure 3: The trace function (center) is used to identify subgoals $g_0$, $g_1$ and $g_2$, corresponding to which policies $\pi_{g_0}$, $\pi_{g_1}$ and $\pi_{g_2}$ are learned. The colors respresent the trace values (for the center image) or scaled value functions (for images corresponding to $\pi_{g_0}$, $\pi_{g_1}$ and $\pi_{g_2}$)

this trace function, we generate subgoals (Algorithm 2) based on the temporal closeness to the goal state. The corresponding subpolicies (visualized in Figure 3) are learned and subsequently used to guide the actions of the agent (Algorithm 3).

As depicted in Figure 3 (a), our approach is effective at obtaining high rewards for these environments where higher rewards are associated with a higher frequency of goal state visits. As seen in the figure, standard $Q-$learning performs poorly in this environment due to its reward-sparse nature. Other approaches such as Episodic Backward Update (EBU) (Lee et al., 2019) performs relatively better, due to quicker credit assignment facilitated by the backward nature of the updates. Prioritized Experience Replay (PER) (Schaul et al., 2016) benefits from sequencing the transitions to be replayed.

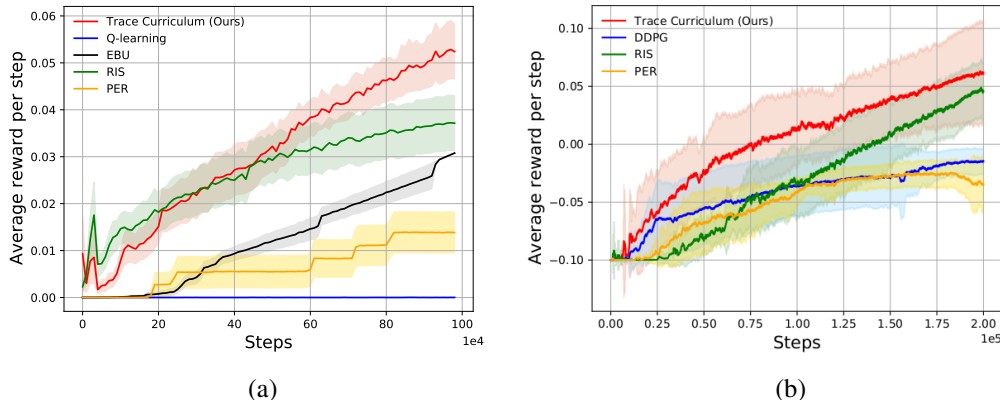

(a)              (b)

Figure 4: Performances in (a) the gridworld environment and (b) the U-Maze environment over 10 and 5 trials respectively.

|  | Trace Curriculum (Ours) | DDPG/Q-learning | RIS | PER | EBU |
|---|---|---|---|---|---|
| Gridworld | **30725** | 3 | 26744 | 6698 | 12334 |
| U-Maze | **18791** | 10403 | 12200 | 8941 | — |
| S-Maze | **1280** | 54 | 1022 | 53 | — |
| $\omega$-Maze | **2287** | 75 | 2090 | 197 | — |
| $\Pi$-Maze | 1262 | 102 | **1273** | 186 | — |

Table 1: Average goal visits across environments and baselines at the end of training for the Gridworld (10 trials) and Point Mujoco Maze (5 trials) environments. Bold numbers indicate best performance.

The approach of Reinforcement learning with Imagined Subgoals (RIS) (Chane-Sane et al., 2021) performs well, as similar to our approach, it also generates a sequence of subgoals that lead towards the goal state. However, our approach benefits from the fact that the subgoal generation is carried out based on the reachability trace function, whose goal-oriented nature, coupled with the backward propagation of the reachability values results in a superior performance.

Similar results can be seen in the Point U-Maze environment, where a DDPG agent (Lillicrap et al., 2016) receives a reward of 1 for reaching the goal state, and a penalty of $-0.1$ for all other transitions. The EBU baseline, designed for discrete action environments, was omitted for the maze tasks. The performance of the agent (in terms of total goal visits) in all the environments and across other baselines is also shown in Table 3. The performance plots for other environments are shown in Appendix J. These results suggest that our algorithm is well-suited to handle sparse rewards, even relative to other competing curriculum learning approaches.

### 5.1 THE CASE OF POORLY DESIGNED REWARD FUNCTIONS

So far, we have discussed the utility of reachability traces in solving sparse reward problems. However, they could also be to useful in specific scenarios such as dealing with poorly designed reward functions (which could lead to the 'couch potato' effect (Burda et al., 2018)). We consider another navigation environment with goal $G$ shown in Figure 5 (a). However, apart from the goal rewards at $G$, the agent also receives receives a small positive reward at the non-terminal state $s_{NT}$. When this reward is set to a high enough value, it distracts reward maximizing agents to learn policies that move towards $s_{NT}$, resulting in a low frequency of goal state visits, which is the actual metric of interest. The 'distractive' reward is assumed to be caused due to mis-specification of the reward either due to human error or due to unforeseen properties of the simulation environment.

In the environment of Figure 5 (a), the agent receives a reward of 5, after which it terminates. The 'distraction' reward for visiting state $s_{NT}$ is set to 0.02, and for all other transitions, the agent receives a living penalty of $-0.5$. With an episode horizon of 200 steps and $\gamma = 1$, strictly reward maximizing

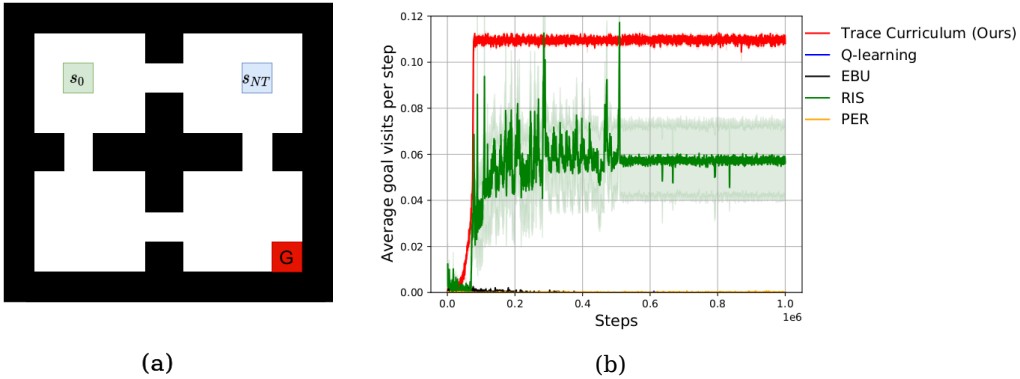

(a)                                             (b)

Figure 5: (a) Environment with starting state $s_0$, goal $G$ and non-terminal rewards at $s_{NT}$, and (b) corresponding performance over $5$ trials.

agents move towards $s_{NT}$ and remain there, receiving a reward of $0.02$ until the episode terminates (optimal episode return$= 3.65$). In contrast to this, the shortest path towards the goal state $G$ would only result in a return of $-1$. This causes reward maximizing agents to become distracted, and learn policies that lead them towards $s_{NT}$. However, with the use of reachability traces, the agent is primarily concerned with task completion (visiting the goal state), and as a result, ignores $s_{NT}$ despite the presence of a distrative reward. As our approach relies on the reachability trace function to generate subgoals, it increases the likelihood that subsequently generated subgoals lie temporally closer to the goal state. This is not the case even in other goal-conditioned approaches like RIS, for example, where subgoal generation is dependent on the agent's value function. As a result, the RIS baseline tends to generate subgoals in high value regions (i.e., around $s_{NT}$) and hence experiences a noisy performance initially. However, RIS being a goal-conditioned approach, eventually discovers paths leading to the goal state. For these reasons, our approach results in faster and more stable learning in the environment in Figure 5 (a), leading to a higher frequency of goal state visits.

## 6   RELATED WORK

Several works have attempted to resolve the sparse reward problem. Reward shaping (Ng et al., 1999) was among the earliest of such approaches wherein the original reward function of the agent was made less sparse by appending it with a special shaping term which left the resulting policy unaffected. Directed and curiosity-based exploration (Thrun, 1992; Burda et al., 2018; Savinov et al., 2018) is another family of approaches designed to speed up learning in general, including in reward-sparse environments. However, these methods are generally designed for better exploration, and not specifically for goal-directed behaviors.

For long-horizon tasks, hierarchical RL (Dayan & Hinton, 1993; Wiering & Schmidhuber, 1997; Levy et al., 2017; Zhang et al., 2021) is a suitable approach, where often, a high level policy controls the execution of several low-level intermediate subpolicies. However, as mentioned in Nachum et al. (2018), the joint learning of these policies can be problematic, and lead to non-stationarity and instability. Although we follow a similar approach of decomposing a goal into multiple subgoals, we use the corresponding subpolicies only to guide the learning of the main policy.

The subgoals generated in our approach is based on the trace function, whose trace values are propagated backwards following a successful trajectory. Lee et al. (2019) proposed similar backward updates in discrete action environments, but the updates were performed on the action value function itself. Although these backward updates were demonstrated to result in faster learning in environments with discrete action spaces, they failed to account for reachability, making them susceptible to failure in environments with poorly designed rewards. Similar to our approach, Savinov et al. (2018) explored the idea of reachability, where a neural network was used to estimate the number of steps separating the current observation from past observations. This prediction was used to generate a bonus reward to encourage exploration into novel regions. Our proposed reachability traces approach,

although based on similar ideas of modeling temporal closeness, is specifically designed to achieve goal-directed behaviors, while also serving as a basis for the choice of subgoal states.

Among curriculum learning approaches, RIS (Chane-Sane et al., 2021) is the most closely related. The authors aim to learn a high-level policy by generating subgoals based on a value function as a reachability metric. The idea is to pick intermediate goal states that are halfway (as per the value function distance metric) to the goal location. Although our approach also relies on a higher-level function to determine subgoals, we choose our subgoals based on the learned trace function, which benefits from the backward propagation of trace values, making it a better choice compared to the value function. Choosing subgoals based on value functions could also fail in cases where the reward function is poorly designed. The trace function, which is based on reachability, does not suffer from such issues. In addition, our subgoal selection strategy is to pick a reachable state with a trace value that is higher than that of the previous subgoal. We believe that this reachability-focused approach is more grounded than the arbitrary criterion of halfway distances.

Other related curriculum learning approaches include Florensa et al. (2018), where subgoals were generated to produce returns between arbitrarily chosen maximum and minimum return thresholds. Apart from subgoal sequencing, algorithms such as prioritized experience replay (Schaul et al., 2016) could also be considered a form of curriculum learning, where the transition samples are sequenced.

## 7  CONCLUSION

We presented reachability trace functions, a novel approach for designing a task curriculum to overcome problems posed by sparse rewards in goal based tasks. We showed how this function approximates the temporal closeness to the goal state, and described how it can be used to construct a task curriculum. Through several discrete as well and continuous maze navigation environments, we empirically demonstrated the ability of our approach to efficiently handle sparse rewards. Finally, we also briefly discussed and empirically demonstrated an alternative use-case of our approach - handling environments with poorly designed reward functions, where reward maximizing agents fail.

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

APPENDIX

## A  LEARNING THE REACHABILITY TRACE VIA REINFORCEMENT LEARNING

Although we described approximating the reachability trace function via a neural network, we note that it could also potentially be learned using a reinforcement learning approach.

**Proposition 1.** *The reachability trace function $\phi(s)$ with a trace decay parameter $\lambda$ and a maximum trace label $e_0$ is equivalent to the optimal value function of an MDP $\mathcal{M}_\phi = (\mathcal{S}, \mathcal{A}, \mathcal{T}, \mathcal{R}_\phi, G, s_0)$ with start state $s_0$, goal state $G$, state space $\mathcal{S}$, action space $\mathcal{A}$, transition function $\mathcal{T}$ and reward function $\mathcal{R}_\phi$ with binary rewards:*

$$r_\phi(s) = \begin{cases} e_0 & s = G \\ 0 & otherwise \end{cases}$$

*and converges to the optimal goal reaching policy with probability 1 under standard convergence conditions $\sum_{t=1}^{\infty} \alpha_t = \infty$ and $\sum_{t=1}^{\infty} \alpha_t^2 < \infty$, where $\alpha_t$ is the learning rate hyperparameter at time step $t$ in the value function update equation for solving $\mathcal{M}_\phi$.*

*Proof.* In RL, the value function $V(s_t)$ associated with a state $s_t$ can be represented as the discounted sum of rewards (as generated by a reward function $\mathcal{R}$) obtained by following a policy $\pi$. That is:

$$V(s_t) = \mathbb{E}_\pi[r_t + \gamma r_{t+1} + ... \gamma^i r_{t+i} + ... \gamma^N r_{t+N}]$$

If $r_T$ is the terminal goal state reward, with all other transitions being associated with a reward of 0, and the agent reaches the goal at step $t + N$, the above relation would simplify to:

$$V(s_t) = \mathbb{E}_\pi[\gamma^N r_T] \tag{4}$$

As the trace function is given by $\phi(s_t) = \mathbb{E}_\pi[\lambda^N e_0]$ (Equation 2), we observe its similarity with Equation 4, and surmise that the optimal reachability function $\phi(s)$ is equivalent to the optimal value function corresponding to a hypothetical MDP with a terminal reward $e_0$ and discount factor $\lambda$. This interpretation allows reachability traces to also potentially be learned via standard RL algorithms. Doing so can guarantee that the learned reachability traces converge to the optimal goal reaching policy (as the binary reward $r_\phi$ is solely associated with reaching the goal state $G$), subject to standard convergence criteria $\sum_{t=1}^{\infty} \alpha_t = \infty$ and $\sum_{t=1}^{\infty} \alpha_t^2 < \infty$ (Jaakkola et al., 1994). As analogous results can be obtained for traces when represented as a function of state-action pairs, this interpretation allows the reachability traces $\phi(s, a)$ of state-action pairs to potentially be learned in an off-policy manner, in parallel with the learning of a policy corresponding to the agent's primary task. We note that learning the reachability trace in this manner would avoid the biases that would otherwise occur when learning the reachability trace. However, it may become infeasible to learn reward sparse environments. Methods such as replaying transitions associated with large rewards (Isele & Cosgun, 2018) could help counter this. □

## B  SELECTING THE SUBGOAL TRACE SCALING FACTOR $t_\phi$

As described in Section 3, our approach selects subsequent subgoals $g_{next}$ by comparing its trace value with that of the most recent subgoal $g_{curr}$ using the condition: $\phi(g_{next}) > t_\phi \phi(g_{curr})$. The scaling factor $t_\phi$ thus must be set to a minimum value of 1 to ensure that the subsequent subgoal corresponds to a higher reachability trace value. Increasing values of $t_\phi$ would tend to produce subgoals that are further apart from each other, causing the total number of subgoals (and subpolicies) to drop in numbers. This effect is observed in Figure 7, where lower scaling factors $t_\phi$ are shown (over 20 trials in the U-Maze environment) to generally produce a larger number of subpolicies (which implies many closely spaced subgoals). Although a lower number of subgoals is desirable, using $t_\phi > 1$ bears a risk of agents being unable to find states satisfying the required reachability trace threshold. In such scenarios, inaccuracies in the approximated reachability trace may get exploited

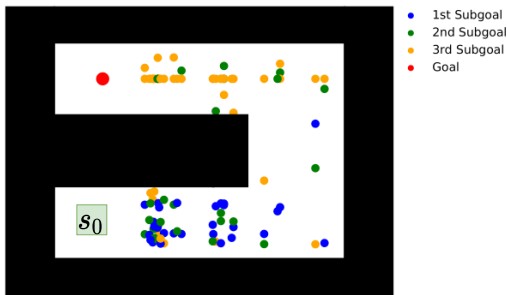

Figure 6: Distribution of subgoals over 30 runs in the U-Maze environment

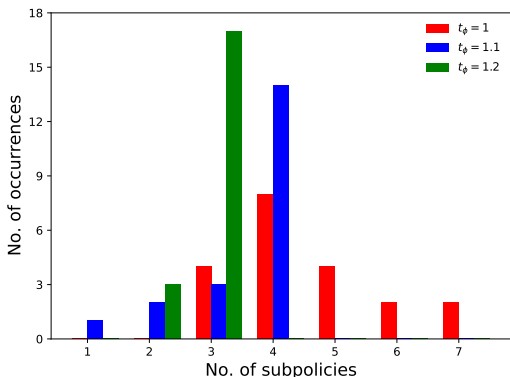

Figure 7: Number of subpolicies learned using reachability traces over 20 runs each for different subgoal trace scaling factor $t_\phi$ in the U-Maze environment

(for example, a subgoal with a high but inaccurate reachability trace could get selected), leading to a poor distribution of subgoals. Hence, despite the potential advantages of using high $t_\phi$, we suggest the use of a scaling factor $t_\phi = 1$.

## C   SENSITIVITY TO STORED TRAJECTORY LENGTH $L$

We ablate the effect of the stored trajectory length parameter $L$ of Algorithm 1 in the environment shown in Figure 2 (a). As observed from Figure 8, increasing $L$ accelerates learning. This probably occurs due to the quicker learning of the trace function owing to training it on longer trajectories.

## D   AVAILABILITY OF DEMONSTRATIONS

As noted in Section 2, our approach relies on there being at least one successful trajectory that reaches the goal state. Hence, our method can also be used in the case of available demonstrations. Such demonstrations need not be optimal, and could be fed in either manually, through other learning approaches or through random exploration. Figure 9 shows the comparison of demonstration based (using a single successful demonstration trajectory) vs demonstration-free versions of our algorithm for the environment in Figure 2 (a). As evident from Figure 9, our algorithm performs marginally better in the presence of demonstrations, owing to the fact that the provided demonstrations are used to construct the trace function as soon as the agent starts interacting with the environment. This is in contrast to the demonstration free case, where the agent is required to first discover successful trajectories through interaction. Nevertheless, our approach can be used either with or without demonstrations. All results reported in the main text correspond to learning without demonstrations.

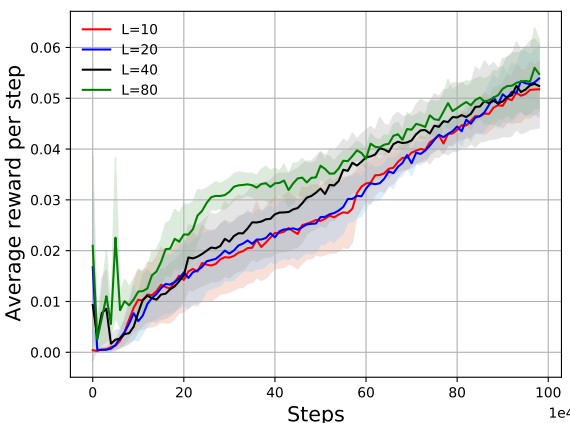

Figure 8: Variation of performance with different stored trajectory lengths $L$ over 10 trials

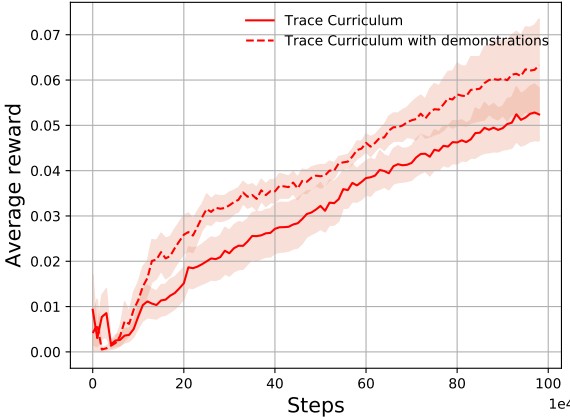

Figure 9: Comparison of our approach with and without the availability of demonstrations over 10 trials in the gridworld environment of Figure 2 (a).

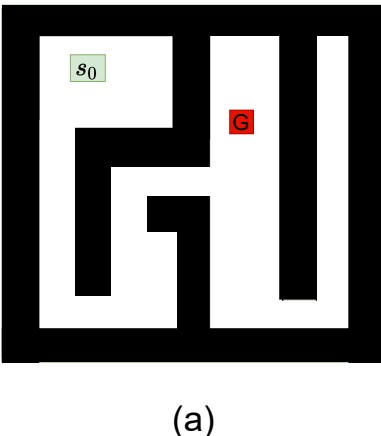
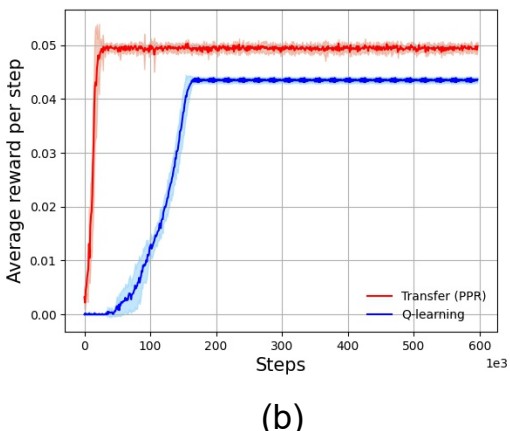

(a)                   (b)

Figure 10: (a) The environment from Figure 2(a) with a new goal location and (b) a comparison of the corresponding performances of standard $Q$-learning vs the policy reuse baseline (PPR), where subpolicies from Figure 3 are reused.

## E   REUSE OF LEARNED SUBPOLICIES

Once our agent uses the trace function to identify and learn subgoals, the corresponding subpolicies are used to guide the agent's goal reaching policy. Although it is not the main focus of this work, these subpolicies could also serve as source policies for a policy reuse scenario. Figure 10 depicts such a case, where the goal state in Figure 2 (a) is changed, with all other environment properties preserved. In such a case, we demonstrate that reusing previously learned subpolicies through probabilistic policy reuse (Fernández & Veloso, 2006) can significantly accelerate learning. Although this type of reuse is possible even with subpolicies learned via other curriculum learning approaches, we believe our method enables the learning of subpolicies whose subgoals are temporally sufficiently apart, making the corresponding subpolicies considerably different from each other. We contend that this could potentially be beneficial for learning (via policy reuse) a wide variety of tasks.

## F   USING REACHABILITY TRACES AS AN EXPLORATION BONUS

Since the reachability traces are essentially indicative of the temporal closeness of a state to the goal state, its value increases for states that lie temporally closer to the goal state. This nature of the reachability trace function can be exploited by using it as an exploration bonus, which can be very effective in sparse-reward scenarios. Figure 11 shows the performance of an agent that whose environment reward at state $s$, given by $r(s)$ is augmented with the trace function $\phi(s)$, which acts as an exploration bonus, such that the total reward is $r(s) + \phi(s)$. As observed from the figure, the agent with the exploration bonus $\phi(s)$ is able to significantly outperform other learning approaches, indicating that the trace function also serves as a useful exploration bonus. We note that the rewards shown in the figure only correspond to the environment rewards obtained by the agent.

## G   VISUALIZATION OF TRACES

Figures 12 and 13 depict the visualization of traces on the point UMaze and in the MountainCar-v0 environment of OpenAI gym (Brockman et al., 2016). As evident from the figures, the trace values increase with proximity to the terminal goal location.

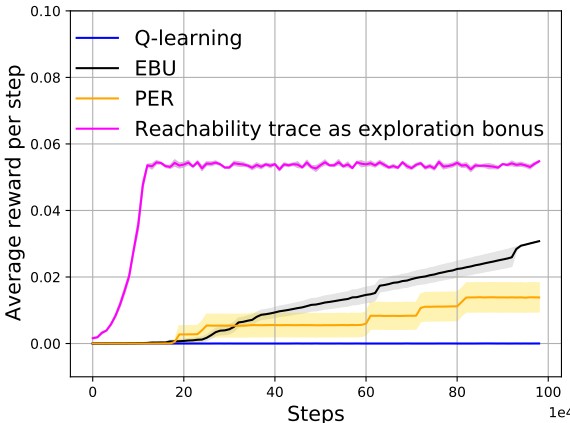

Figure 11: The performance of reachability traces used as an exploration bonus, compared to other approaches averaged over 10 trials in the gridworld environment of Figure 2 (a).

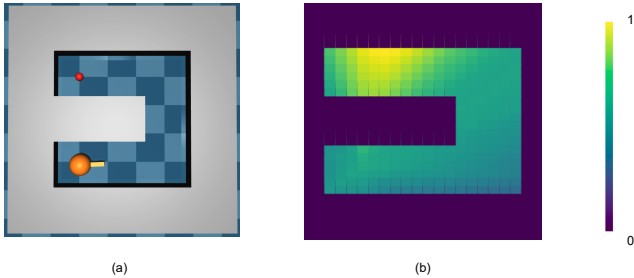

Figure 12: (b) Visualization of traces in the (a) UMaze Environment

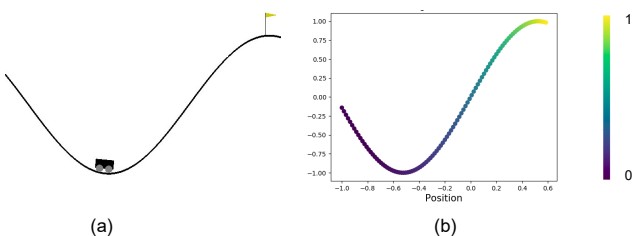

Figure 13: (b) Visualization of traces in the (a) Mountain Car-v0 environment

| Hyperparameters | Gridworld | Point U-Maze | Other Point mazes |
|---|---|---|---|
| learning rate $\alpha$ | 0.05 | $1e-5$ | $1e-5$ |
| discount factor $\gamma$ | 0.95 | 0.95 | 0.95 |
| Episode horizon $H$ | 200 | 500 | $50K$ |
| Total steps | $1e6$ | $2e5$ | $1e6$ |
| Stored trajectory length $L$ | 40 | 50 | $2K$ |
| Size of trace buffer | 500 | 300 | $10K$ |
| Subgoal trace scaling factor $t_\phi$ | 1 | 1 | 1 |
| Epochs for training $\phi$ | 100 | $10K$ | $10K$ |
| Trace decay parameter | 0.9 | 0.99 | 0.9999 |

Table 2: Hyperparameter values for our approach.

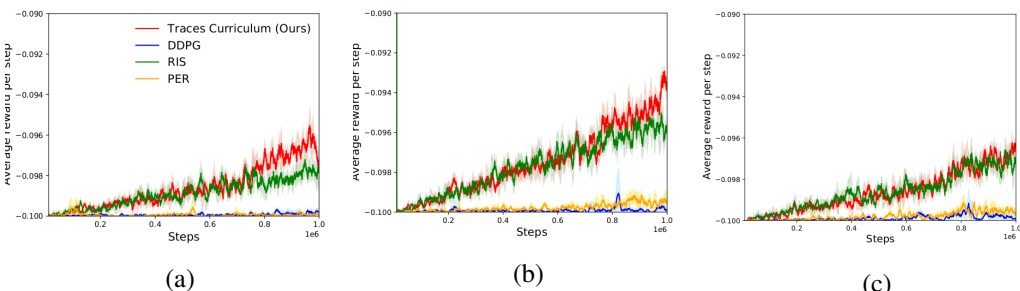

Figure 14: Performances in the point (a) S-Maze, (b) $\omega$-Maze and (c) $\pi$-Maze environment overs over 5 trials.

## H  HYPERPARAMETER SETTINGS

In these environments, the exploration parameter $\epsilon$ decayed linearly from 1 to a minimum of 0.1. The DDPG base agent used in the continuous maze tasks used the Adam optimizer (Kingma & Ba, 2014) and a critic and target networks as multi-layer perceptrons of 3 and 2 layers respectively, with the former having 1024, 512 and 300 nodes in its three layers, and the latter with 512 and 128 nodes in its layers. The polyak averaging hyperparameter used was set to $\tau = 0.01$. For the PER baseline, we used proportional prioritization with a temperature parameter of 0.01. Each continuous maze environment was performed on an Nvidia Tesla V100 (32GB) GPU, and on average, took about 6 hours per trial for U-Maze and about 12 hours per trial for the other continuous mazes.

The trace network used is a feedforward neural network with two hidden relu layers, each containing 128 nodes. The optimizer used is Adam with a learning rate of $1e - 3$.

## I  OTHER IMPLEMENTATION DETAILS

*Efficient Trace Updates:* As per the procedures outline in Algorithm 1, we train the trace network at each step. However, this may become prohibitively expensive, and thus impractical in terms of wall-clock time. In order to counter this issue, we keep track of the latest interaction that led to the goal state, at which point we set a spiking signal $\zeta$ to 1. This spiking signal is then subsequently decayed exponentially as $\zeta_{t+1} = d\zeta_t$, where $d$, the decay rate is set to 0.9. The trace network is then updated with a probability $\zeta$, such that immediately after a goal state visit, the trace network is updated frequently, following which it is updated progressively less frequently.

*Off-policy updates:* As our learning agent is updated in an off-policy manner, its action values can simultaneuosly be updated in parallel during the environment interactions in Algorithm 1 and 2, and not just in Algorithm 3.

*Curriculum guidance:* In Algorithm 3, although we use a fixed value of $\epsilon = 0.1$, as per standard practice, $\epsilon$ could be set to decay with time, decreasing the reliance on subpolicies with more learning updates. A potential drawback of this strategy is that the agent may receive less frequent advice from subpolicies further down the sequence. However, the agent, having been guided by subpoliciies earlier in the sequence, is likely to be closer to the goal state at this stage, and is thus less likely to be affected by the low frequency of advice.

*Baseline implementation:* For a fair comparison, in all our experiments, the RIS baselines only used the subgoal generation strategy from Chane-Sane et al. (2021) without the use of hindsight experience replay (Andrychowicz et al., 2017).

## J  OTHER PERFORMANCES

Figures 14 (a)-(c) show the performance plots in other point Mujoco maze environments (S-Maze, $\omega$-Maze and $\pi$-Maze).

|  | Trace Curriculum (Ours) | DDPG/Q-learning | RIS | PER | EBU |
|---|---|---|---|---|---|
| Gridworld | $\mathbf{0.052 \pm 0.006}$ | $1e-5 \pm 1e-5$ | $0.037 \pm 0.006$ | $0.014 \pm 0.004$ | $0.031 \pm 2e-5$ |
| U-Maze | $\mathbf{0.061 \pm 0.004}$ | $-0.014 \pm 0.011$ | $0.046 \pm 0.024$ | $-0.035 \pm 0.024$ | $-$ |
| S-Maze | $\mathbf{-0.0973 \pm 3e-4}$ | $-0.0999 \pm 5e-4$ | $-0.0976 \pm 6e-4$ | $-0.0998 \pm 9e-5$ | $-$ |
| $\omega$-Maze | $\mathbf{-0.0937 \pm 7e-4}$ | $-0.0999 \pm 9e-5$ | $-0.0958 \pm 2e-4$ | $-0.0995 \pm 2e-4$ | $-$ |
| $\Pi$-Maze | $\mathbf{-0.0964 \pm 5e-5}$ | $-0.0999 \pm 1e-6$ | $-0.0976 \pm 2e-4$ | $-0.0997 \pm 2e-4$ | $-$ |

Table 3: Average reward per step (mean±standard deviation) across environments and baselines at the end of training for the Gridworld (10 trials) and Point Mujoco Maze (5 trials) environments. The bold values indicate the best performance. The averaging window size used for U-Maze is 1000, and for other mazes, it is 10000.

