# OpenReview forum: "Reachability Traces for Curriculum Design in Reinforcement Learning"
_ICLR.cc/2022/Conference — ICLR 2022 Submitted_

### Official Review · Reviewer_psX9 · 2021-10-28

**Correctness:** 3
**Technical Novelty And Significance:** 1
**Empirical Novelty And Significance:** 2
**Recommendation:** 3
**Confidence:** 4

**Main Review:**

This paper is one of the papers that try to automatically propose a curriculum learning to improve the sample efficiency of the RL algorithms. Similar ideas has been intensively explored by many papers. See a new review paper https://arxiv.org/pdf/2003.04664.pdf. The algorithm this paper proposed has a strong limitation. It only works when there is one final goal. If the goal of the learning is to train an agent that can reach multiple different goals, one has to learn the trace function multiple times. I also have the following concerns.

1. The proposition 1 is not clear. I am not sure what it means by "the reachability trace function is a solution to an MDP". And when the author say "converges to the optimal goal reaching policy", which specific algorithm are we talking about? Is it supposed to be $\mathcal{R}_{\phi}$ instead of $r_{\phi}$ in Proposition 1?  By reading the proof of Proposition 1, it is very vague. It is not clear what the authors mean by "This interpretation allows reachability traces to also potentially be learned via standard RL algorithms."

2. I don't agree with the statement at the start of section 3. When $\phi(s)$ is high, it may be less reachable from the starting state. However, $\phi(s)$ is low, it only indicates that $s$ is far from goal state, it does not necessarily mean it is close to the starting state. For example, in a navigation problem, your starting state is in the middle of the map and goal state is on the right corner. Then any state on the left corner could also have a low $\phi(s)$ while they are not good subgoals.

3. Algorithm does not show a significant improvement over RIS. The authors should compare to more baseline algorithms. At least, HER (hindsight experience replay) is an important method for multi-goal learning and it should be compared.






**Summary Of The Paper:**

This paper proposed an automatic curriculum learning design method for goal-orientated Reinforcement Learning. The idea is to learn a reachability trace function to measure the temporal difference between a state and the final goal state. Then a curriculum is learnt based on the reachability trace function. The algorithm is evaluated on tabular and continuous robotic control environments.

**Summary Of The Review:**

I believe the idea behind the paper is not very novel and the performance is not very significant. The theory is poorly written. I recommend rejection.

---

> ### Author Response · Authors · 2021-11-22
> **Responses to Reviewer psX9**
>
> We thank the reviewer for their comments. We have taken the review paper into account and cited it in the revised draft. It is true that the approach learns to navigate to just a single goal. However, this is also the case in other recent curriculum learning approaches, which do not consider the multigoal case. We leave the integration of our approach with multigoal approaches for future work.
>
> $\textbf{The proposition 1 is not clear:}$ We recognize that the term 'solution to the MDP' lacks clarity. Our intention was to state that the learned trace function corresponds to the learned value function of the hypothetical MDP $\mathcal{M_\phi}$.
> We have corrected this in the revised draft. In the proposition, $\mathcal{R_\phi}$ is the reward function corresponding to $\mathcal{M_\phi}$, and ${r_\phi}$ is the instantaneous rewards received during learning. By the possibility of the traces being learned via standard RL algorithms, we meant that since the trace function is essentially the value function of a hypothetical MDP $\mathcal{M_\phi}$, it could also be leared via reinforcment learning algorithms, instead of via neural networks. We have also moved the proposition to the Appendix, as we believe it stands alone, and is less distractive compared to including it as part of our main text.
>
> $\textbf{I don't agree with the statement..:}$ The closeness we are referring to here is the temporal closeness and not the spacial closeness. Due to the manner in which the trace labels are assigned (they are assigned a high value for states temporally closer to the goal, and low values for states temporally further away from the goal), for a given successful trajectory, the lowest trace value occurs at the start state, which is the temporally the furthest away from the goal state. We have clarified this point in the revised draft.
>
> $\textbf{Algorithm does not show..}$ Section 5.1 depicts specific scenarios in which our algorithm is superior to other baselines such as RIS. In addition, empirically, we show that our approach performs better or at least on par with RIS. This is due to the fact that RIS determines subsequent subgoals based on the value function, which is usually slow to converge. In contrast, the subgoals generated by our approach are based on the trace function, which is learned relatively much faster, owing to the fact that the trace labels associated with multiple states within the trajectory are used to update the trace function.

---

> > ### Comment · Reviewer_psX9 · 2021-11-25
> > **Thanks for the response**
> >
> > The authors response does help me understand the paper better. However, I still doubt the significance of the paper. The theoretical analysis is very weak and the authors decided to move Proposition 1 to the Appendix. The algorithm design is very complicated while the environments that the paper tested are very limited to the navigation problem. I doubt the generalization ability of the proposed approach.
> >
> > I decide to keep my original rating for the paper.

---

### Official Review · Reviewer_8Hsg · 2021-11-02

**Correctness:** 1
**Technical Novelty And Significance:** 3
**Empirical Novelty And Significance:** 1
**Recommendation:** 3
**Confidence:** 4

**Main Review:**

* Pros
  * The idea of discovering a sequence of subgoals and using it as a curriculum is interesting.

* Cons
  * The notion of reachability is not particularly novel.
  * The proposed method is not technically sound and requires several strong assumptions.
  * Lack of discussion and comparison to related work.
  * The empirical results are not convincing.

---

- The notion of reachability function is not much interesting because it is equivalent to value function with a particular reward function (i.e., 0 for all non-goal states and $e_0$ for goal state) as acknowledged by the paper. It seems like the underlying claim of this paper is that it is useful to use the particular reward function for goal-based RL, which is in fact a standard setting in goal-based RL. Instead of claiming a new concept, it would be clearer to say that this paper shows how this particular reward function can be used for curriculum design in goal-based RL.

- There is a large body of prior work on dealing with sparse rewards in goal-based reinforcement learning (e.g., HER and its follow-up work). This paper does not discuss how this work is related to the prior work. The experiments are also missing comparison to the related prior work.

- The proposed algorithm does not seem technically sound and requires strong assumptions for the following reasons.

  1. Algorithm 1 for learning reachability function seems inconsistent with Equation 2, because the data for training the reachability function is sampled from successful trajectories and unsuccessful trajectories with **equal probability**. However, the data needs to be sampled according to the policy $\pi$ in order to estimate Equation 2. In other words, Algorithm 1 would lead to a biased estimation of Equation 2.

  2. The proposed method requires a randomly-initialized policy to generate at least one successful episodes even before the proposed algorithm is applied. Thus, the proposed algorithm is restricted to relatively easier sparse-reward problems unlike the prior work (e.g., HER) that is expected to work in harder settings, where random exploration does not reach the main goal.

  3. The paper says "a low $\phi$ value may indicate that the state is further away from the goal state (and perhaps closer to the starting state), and thus may consitute an easier subgoal to be learned." This seems to assume that the starting state is the farthest state from the goal state, which is not necessarily true. ​Imagine a length-N chain MDP where $s_i$ is connected to $s_{i-1}$ and $s_{i+1}$. If the goal state is $s_N$ and the initial state is very close (say $s_{N-1}$), the reachability function of the initial state ($\phi(s_{N-1})$) would be very high, while the $\phi(s_0)$ would be very low. In this case, the proposed algorithm would learn to reach the farthest state ($s_0$) as the first subgoal because it has the lowest $phi$ value. It would is not only difficult to learn but also could hinder learning about the main goal. Although this is an adversarial example, the paper needs to clarify this underlying assumption.

  4. Algorithm 3 goes through the sequence of discovered subgoals and moves on to the next subgoal whenever the current goal is reached by the agent. This seems to assume that subgoals are on the optimal path towards the main goal. Otherwise, if the subgoal is not a part of the optimal path towards the final goal, the subpolicy corresponding the current subgoal could hinder learning by choosing actions that lead to states that are even farther away from the main goal state. It is important to either show that the assumption holds or propose a way to overcome this issue.

- The experimental results are not convincing.
  1. Does the x-axis in Figure 4 include the amount of data used for Algorithm 1 (reachability learning) and Algorithm 2 (subgoal learning)? If not, it is not a fair comparison. If yes, it is important to show how much data is used for each step (algorithm 1, 2, 3) and how they are determined.
  2. The experiment design for poorly designed reward function (Section 5.1) does not sound reasonable. According to the reward structure, it is optimal to stay at the $S_{NT}$ rather than reaching the goal state. It is a bit unreasonable to compare different algorithms under different reward functions (e.g., baselines given the poorly designed reward function and the proposed method given a good reward function). This result shows nothing more than saying that the reachability reward function (i.e., 0 for all non-goal states and a positive reward for goal state) is a reasonable reward design for goal-based RL.

**Summary Of The Paper:**

This paper considers a sparse reward goal-based RL problem and introduces a notion of reachability trace for reinforcement learning, which intuitively approximates how reachable (or how far) the goal state is from the current state. The proposed idea is to divide the whole learning problem into three phases: 1) learning a reachability function w.r.t. the main goal state, 2) finding a sequence of subgoals that are gradually closer to the main goal, 3) learning a main policy while using the sequence subgoals (corresponding subpolicies) as an exploratory policy in epsilon-greedy action sampling. The empirical results on grid worlds and several maze environments show that the proposed method outperforms DDPG, RIS, PER, and EBU.

**Summary Of The Review:**

Both exploration in sparse-reward goal-based RL and subgoal discovery are important topics in RL. Although this paper contains an interesting idea for subgoal discovery and curriculum design, the proposed algorithm requires several strong assumptions does not technically sound, and the empirical results are not convincing enough and lacks comparison to important baselines.

---
**Update after the rebuttal**

I'm not entirely convinced by the author's rebuttal for the following reasons. Thus, I will keep my original score.
1. If the reachability learning algorithm is biased and not consistent with Proposition 1, I do not see why Proposition 1 is needed from the first place.
2. The author's justification for not comparing against HER is not convincing. Single-goal RL is just a special case of multi-goal RL. In fact, HER has been shown to be effective in a single-goal RL setting (see Section 4.3 of the HER paper). So, I believe that it is important to compare against this baseline.
3. The authors should have clearly showed how much data each algorithm (1, 2, 3) consumes in the plots and how the switching points are determined. The fact that the total amount of data is reflected in the x-axis doesn't address my concern.
4. I'm still not sure why Section 5.1 is needed. My initial comment about "This result shows nothing more than saying that the reachability reward function (i.e., 0 for all non-goal states and a positive reward for goal state) is a reasonable reward design for goal-based RL." is not addressed by the author's response. The author's response seems to repeat exactly what I wrote above. This paper seems to propose a proper reward design (reachability) as a solution, but my comment was that a proper reward design seems to be on the problem side rather than on the solution side.

---

> ### Author Response · Authors · 2021-11-22
> **Responses to Reviewer 8Hsg**
>
> $\textbf{The notion of reachability function is not much interesting...}$ It is true that the trace function is equivalent to another value function, as you have rightly mentioned. However, unlike standard RL algorithms, the fact that the reachability values of the trajectory is used (as opposed to the reward of just a single state) allows this function to be learned much faster than value functions learned through standard approaches like Q learning.
>
> $\textbf{There is a large body of..}$ We agree that HER-inspired methods address sparse reward, goal-based reinforcement learning problems. However, these methods are inherently multi-goal approaches based on Universal Value Function (UVFs), where the input depends on states as well as goals, and can be expensive to train. Moreover, we focus on proposing an approach to design a curriculum of tasks, each of which could be learned using single or multi-goal approaches like HER. We leave this integration for future work.
>
> $\textbf{Algorithm 1 for learning reachability..}$ We thank the reviewer for pointing this out. It is indeed true that Algorithm 1 learns a biased version of Eq 2. However, in reward-sparse scenarios, such a biased estimate can still be useful, as it provides some form of signal to indicate temporal proximity to the goal. We do not claim equivalence of the trace function learned in Algorithm 1 and the one in the proposition. To make this clear, we have moved the proposition (which stands alone) to the Appendix.
>
> $\textbf{The proposed method requires..}$ We distinguish our work from HER-related works as the main focus is to design a curriculum of tasks in goal-based environments. We recognize that extending our approach to multigoal scenarios is important, but we leave it for future work, and instead focus on the problem of generating curricula for goal-based tasks.
>
> $\textbf{The paper says ''a low $\phi$ value..}$  For a given successful trajectory, the trace labels are assigned such that they always have higher values closer to the goal and lower values closer to the initial states. The closeness being referred to here is temporal closeness and not spacial closeness of the states. We have clarified this in the revised draft.
>
> $\textbf{Algorithm 3 assumes..}$ There is no assumption regarding the optimality of the subgoal sequence, as discussed in Section 4. The sequence of subgoals is only designed for the agent to experience goal state visits more frequently, but not necessarily in an optimal manner. The off-policy nature of the learning algorithm used ensures that the optimal policy learned even when the actions chosen are sub-optimal.
>
> $\textbf{Does the x-axis..}$ In all figures, the performance curves for our method are not only for Algorithm 3 - they include the environment interactions for Algorithms 1 and 2 as well. The initial phase of learning is controlled by Algorithms 1 and 2, and once a curriculum is learned, Algorithm 3 takes over.
>
> $\textbf{The experiment design for..}$ It is true that staying at $s_{NT}$ is optimal as per the given reward function. However, we contend that designing good reward functions are often not easy, and incorrectly designed reward functions lead to unexpected behaviors. Section 5.1 was designed just to show that in such cases, our algorithm would still be able to reach the goal state, as its design is based solely on reachability. The assumption here is that the actual intention of a user is to get an agent to navigate to the goal location, and that the non-terminal reward is an outcome of inaccurate reward design or unforeseen properties of the simulation environment. Hence, it would remain un-distracted by other inaccuracies in the reward function, as long as the terminal goal state is specified correctly.

---

### Official Review · Reviewer_zxez · 2021-11-03

**Correctness:** 3
**Technical Novelty And Significance:** 3
**Empirical Novelty And Significance:** 2
**Recommendation:** 5
**Confidence:** 5

**Main Review:**

The high level idea proposed by the paper is very intuitive and interesting — use an initial suboptimal policy to find a (suboptimal) path to goal from all states in the environment, find a subgoal sequence using this coarse map, and use it to build a curriculum that eventually learns to reach the goal. The idea of learning the reachability traces is quite nice and I believe it can be very useful in many contexts. However, I have concerns regarding the purpose of the proposed method (and hence the contributions) and the empirical evaluations. I list my concerns below:

1. [On the overall method] It is not clear to me what the entire subgoal generation + curriculum building exercise “gets us”. Looking at the choice of hyperparameters, [Appendix A] setting $t_\phi=1$ is practically saying that the subgoal generation process is simply leading the agent in direction of the goal as per an increasing $\phi$ trajectory. I am wondering if the authors tried using $\phi$ simply as an exploration bonus without doing any curriculum design. Having learned the reachability traces somewhat gives you a “densification” of the reward, with the catch that it gives you a densification of a suboptimal trajectory. My hunch right now is that this along would be a pretty useful signal to guide the agent towards the goal (which is also what the subgoal sequence is doing) without having to involve so many steps. My general unhappiness with this is that there’s a lot of moving parts that would need tuning and careful reimplementation for new environments/tasks, and it feels over-engineered.
2. [On empirical evaluation] When doing the empirical evaluations, do the other baselines (RIS, Q-learning etc.) have access to the behavior policy/demonstrations used by the the proposed method in Alg 1 and Alg 2? The performance curves seem to only be about the Alg 3 part of the story and if that is the case, the performance is a bit unfair. Bootstrapping any of the baselines with a pool of initial data can greatly accelerate the performance.
3. [On “contributions”] I am not entirely convinced that the contributions [Section 1] are fairly evaluated in the experiments. The paper would greatly benefit by a set of explicit ablation studies that evaluate the usefulness of (say) the reachability traces in isolation from the proposed framework. One way to do this would be, as suggested above, just use the learned $\phi$ as a signal for guiding the agent, and just use the subgoal proposal framework in isolation with some other dense metric for generating subgoals (e.g. IRL rewards etc.). This goes back to my concern about not being convinced whether the multiple moving parts of the algorithm are actually beneficial.
4. This is less of a concern with the paper and more of a discussion — the learning of $\phi$ in Section 2 seems very close to doing IRL using the behavioral policy/demonstrations/offline data. If a connection can be made, a lot of the interesting ideas from the several decades of progress in IRL can be used to improve what $\phi$ can mean, and perhaps, be useful to the overall performance (or standalone). A discussion by the authors (response and in the paper, for the general quality of prose) would be quite useful.
5. [On evaluation environments] The simplistic nature of tasks considered in the empirical evaluations — the gridworld and 2D navigation tasks — is a bit unappealing to me and it is unclear if the insights drawn from these tasks is reflective of the broader set of environments of interest in the RL community. While this is mostly a choice the authors are free to make, I would strongly urge them to include results from a broader set of tasks to make the results appealing to a wider community.
6. [On the nature of tasks] The authors address the task of curriculum generation for single-goal RL, and that feels a bit restrictive, and extensions of the proposed method to the general goal-conditioned, or multi-task, RL framework is not obvious to me. This, coupled with the choice of environments evaluated in, makes the prospects of future directions or applications in complex, real-world domains a lot harder to follow. If it’s something the authors have thought above, the paper would really benefit from a discussion on how reachability traces may be extended to multi-task/goal-conditioned cases (e.g. using hindsight relabeling or a tuple of reachability traces, or a reachability map that maintains arbitrary A to B connections).



I also have another small concern that does not affect my review but would be nice to have some clarification on:
7. [Figure 3] Is this figure showing a learned reachability map (for the center fig) or a visualization using an oracle map? Either way, I am a bit confused as to why value drops in the bottom right corner but is still high in the 3rd column from the right. Any path taken by a behavioral policy can not have crossed the black walls and hence, the middle of the grid (despite being closer to the goal in grid difference) should be farther away than the bottom right corner, similar to what the figure for $\pi_{g_2}$ shows. Is this a mistake or am I missing something?

---

*Update*: Updating my recommendation to reflect the discussion by the others.

**Summary Of The Paper:**

The paper tackles the problem of curriculum design for single-task/single-goal reinforcement learning problems with sparse rewards. The primary contribution of the work is a method to generate a subgoal curriculum using reachability traces — a learned metric that captures the distance to goal using under a pre-determined behavioral policy. The recipe can be summarized as follows: (i) Use an _available_ behavioral policy $\pi$, or demonstrations, to obtain interactions from the environment, some of which reach the goal; learn a reachability function $\phi(s)$ that creates a connectivity map, (ii) generate a sequence subgoals using this connectivity map, eventually leading up to the final goal G, (iii) use the subgoal curriculum to learn to reach goals.


**Summary Of The Review:**

The paper presents an interesting idea of using reachability traces that is implemented with a lot of moving parts and evaluations on simplistic domains. I have some concerns regarding the empirical evaluations and the usefulness of the entire approach and look forward to engaging with the authors in the discussion period.

---

> ### Author Response · Authors · 2021-11-22
> **Responses to Reviewer zxez**
>
> $\textbf{1. On the overall method:}$ It is true that the reachability trace function can be used as an exploration bonus. We have added a section in the appendix to show this performance in the environment of Fig 2(a). As observed, the performance of the agent with the exploration bonus is much superior to an agent without such a bonus. We list this as an additional benefit of learning the trace function. However, the main aim of the current work is to design an algorithm for automated curriculum design, for which Algorithms 2 and 3 are required. One possibility is to use the trace function as an exploration bonus during the subgoal learning process, which will likely make the learning more efficient. However, we leave this for future work. Nevertheless we thank the reviewer for their useful insight.
>
> $\textbf{2. On empirical evaluation:}$ We point out that none of the evaluated approaches (including ours) initially have access to demonstrations (except in Fig 9 in Appendix D, where we show the case of learning with the availability of demonstrations). They all learn from scratch. The performance curves for our method are not only for Algorithm 3 - they include the environment interactions for Algorithms 1 and 2 as well. The initial phase of learning is controlled by Algorithms 1 and 2, and once a curriculum is learned, Algorithm 3 takes over.
>
> $\textbf{3. On ``contributions":}$ Fig 6 and 7 in the appendix aim to study the subgoal generation process in isolation from the other components of our approach. As previously mentioned, we have included a section in the appendix showing the effectiveness of using the trace function as an exploration bonus.
>
> $\textbf{4. This is less of a concern..:}$ We are not entirely sure in what sense the learning of a trace function is related to inverse reinforcement learning (IRL). We note that our approach does not assume access to demonstrations, although it is also compatible when demonstrations are available (Appendix C). We view it as being more similar to a regular reinforcement learning process with backward updates (the EBU baseline). A brief discussion on this relation can also be found in the related works section.
>
> $\textbf{5. On evaluation environments:}$ We selected the environments in Fig 2 (b)-(e) as these were the continuous environments chosen in recent related papers, Chane-Sane et al. (ICML 2021) and Florensa et al. (ICML 2018). We introduced the environment in Fig 2 (a) in order to conduct simple experiments in a classical setting with discrete states and actions, where action values need not be approximated by neural networks.
>
>  $\textbf{6. On the nature of tasks:}$ We agree that extensions to multi-task frameworks and more general settings is important. HER-like methods methods are inherently multi-goal approaches based on Universal Value Function (UVFs), where the input depends on states as well as goals, and can be expensive to train. Moreover, we focus on proposing an approach to design a curriculum of tasks, each of which could be learned using single or multi-goal approaches like HER. We leave this integration for future work.
>
> $ \textbf{7. Figure 3:}$ The center figure in Fig 3 is a visualization of the reachability trace function. As the trace function is just an approximation of the reachability values, it is sometimes possible for it to be poorly approximated (the trace function depicted in the Figure was extracted from a very early stage of learning, which is why it has only partially converged), which is why lower values are seen on the bottom right. We have updated this with a figure that is more representative of what the trace function (from a later stage of learning) should look like.
>  The other figures (non-center figures) in Fig 3 are just visualizations of the value function scaled to the range [0,1]. These depict the value functions learned for the different subgoals ($g_0$, $g_1$ and $g_2$) identified by the trace function. We will include this information our revised draft. Thanks for pointing this out.

---

> > ### Comment · Reviewer_zxez · 2021-11-25
> > **Thank you for the response**
> >
> > Thank you for the detailed responses and running new experiments using FR for exploration. I am bumping up my score by 2 points to reflect this, since it does paint a slightly better picture of what the representation is capable of doing.
> >
> > However, I still have concerns (as highlighted in the main review) regarding the significance of the contributions and the thoroughness of empirical evaluation. I share these concerns with reviewers WVWS and 8Hsg and hence stand by this recommendation.

---

### Official Review · Reviewer_WVWS · 2021-11-03

**Correctness:** 2
**Technical Novelty And Significance:** 3
**Empirical Novelty And Significance:** 2
**Recommendation:** 3
**Confidence:** 3

**Main Review:**

Using the Neurips rubrics:

Originality: To me the idea of using reachability traces seemed reasonably novel, but not a stunning new insight. It is a natural follwup to the idea of using the value functions to define sub-goals in the Chane-Shane paper, but connecting to reachability traces in the old RL literature is a non-trivial contribution. I am not an expert on the latest work in CL in RL so can't judge from that perspective. The authors did seem to survey the related work carefully and situated their work well within it as being a new approach. One minor novelty in the experimental section was the idea of experimenting with misspecified reward function.

Quality/Correctness:  There are 2 problems:

a. Proposition 1 just seems wrong to me and the statement of it kind of meaningless. The reachability trace is a function of the policy $\pi$  and it's claimed to be equivalent to a "solution" (optimal value function?) of an MDP, without any policy specified [also what does "converges to" mean in this context? what sequence converges? ] . This can't be correct and looking at the proof, I think the error is clear.  The authors observe that 2 equations are "similar" but they are actually referencing 2 different policies, 1 the original policy and the other is the optimal policy for this MDP. The expectations will be different. The best that you can say is that the trace function is the value function for the exploratory policy $\pi$. The rest of the discussion in the "proof" isn't really a proof  but a discussion which doesnt apply anymore since we aren't talking about an optimal policy.

b. I think the kind of tasks that this is being applied to are making it much easier for this CL method. In all the environments there is "topologically" speaking one path from start to goal. there are no loops to choose between. So what this means is that the first reachability trace path will lay out roughly the right path to the goal. In loopy environments I suspect you will find more cases of CL hurting. Fig 5a has a loop, but there's only one, and it looks very symmetrical so either path would lead to a good goal policy.

c. The thoroughness of experiments is maybe slightly below par for an ICLR paper. I would have liked to see a little more variety in problem types (rather than just gridworld), also a few more variations e.g. what happens if you use more than one trace at a time?

A counter to the argument from authors that their method works better when rewards are mis-specified: wouldn't RIS then work better when rewards are well-specified?

Clarity: Mostly clear. One clarification I would appreciate is the perceived role the reward function in a setting like this. If you have a goal, but a reward function that is "mis-specified", what does that mean? Ordinarily the reward function by definition is the source of truth for the task. So perhaps the authors are conceiving of reward functions constructed by reward-shaping or something similar?
   Eq is kind of misleading. It looks like the expression in the brackets is a constant until you realize it actually is a function of s_t and is the variable that the expectation is taken over. $\phi$ is originally defined (declared?) as relative to $\pi$ but that dependence on $\pi$ is dropped later, which is confusing.




**Summary Of The Paper:**

The paper introduces a new method of doing curriculum learning for goal-directed RL. It uses a notion of reachability traces to model "temporal closeness" of states to a goal state. The reachability trace function is then used to define a sequence of sub-goals for which policies are learned iteratively, and then the final policy is learned  using advice from the decomposed sub-policies.

Experiments are run on a few gridworld like experiments showing favorable performance vs other CL methods, it is also shown that the method may be more robust when reward function is poorly misspecified.

**Summary Of The Review:**

An interesting, somewhat novel approach, but experiments not thorough enough to show that method is truly robust; theoretical contribution seems completely wrong.

---

> ### Author Response · Authors · 2021-11-22
> **Responses to Reviewer WVWS**
>
> $\textbf{Proposition 1 seems wrong..}$ As rightly pointed out, the reachability trace function is a function of the policy. By 'solution', we do mean the optimal value function, but pertaining to a hypothetical MDP with a modified reward structure (the one mentioned in Proposition 1). What this means is that the learned trace function would correspond to the optimal value function of this hypothetical MDP. It is the convergence of this value function that is being referred to. We do not actually refer to the optimal policy when computing the expectations. Our message is basically that the optimal trace function is equivalent to the optimal value function of the hypothetical MDP. In both equations, $\pi$ refers to the behavior policy (and not the optimal policy) being executed.
> We concede that terms such as `solution' impede clarity, and causes confusion. We have modified our draft accordingly. We hope this addresses your concerns regarding the correctness. We have also moved the proposition to the supplementary material, as we believe it can stand alone, and is less distractive there, compared to including it in the main text.
>
> $\textbf{I think the kind of...}$ Our work specifically addresses goal-based tasks for which the chosen environments or similar environments (barring the one in Fig 2(a), which is a simple grid world environment that we introduced) have been previously used in other recent curriculum learning papers (Chane-Sane et al. (ICML 2021), Florensa et al. (ICML 2018)). We contend that the 'loopy' environments would still not pose a problem, as the reachability of a state is dependent on its temporal (and not spacial) proximity to the goal state. In theory, given an adequate number of successful trajectories, the reachability trace function would provide a reasonable approximation of the number of steps to the goal (temporal proximity).
>
> $\textbf{The thoroughness of the experiments...}$ As mentioned in our previous point, we consider goal-based tasks for which, the navigation environments used in our work seem to be well suited. In addition, the use of these environments in previous related works has influenced our choice for using it in our work as well. Nevertheless, we agree that a more diverse set of environments would be useful. We are unsure about the context in which you mention the use of multiple traces at a time, so we are unable to comment on it.
>
> $\textbf{A counter to the ...}$ In cases where the rewards are well-specified, RIS does work well, as indicated in the experiments section. However, as RIS relies on the estimated value function to determine subgoals, the accuracy of the subgoals is reliant on how well the value function has been approximated. We argue that reachability traces are faster to learn, owing to the backward nature in which the trace labels are assigned. For example, in a tabular environment, after the goal state is visited, the trace labels of multiple states would be updated. In contrast to this, an approach to update the value function would tend to update the value of just the state leading to the goal state. Consequently, learning the trace function is much faster than learning a value function, due to which, our curriculum generation approach is able to identify potentially useful subgoals faster than an approach like RIS, which relies on the value function estimate.
>
> $\textbf{One clarification I would appreciate..}$ It is true that the reward function is in most cases, considered the source of truth of the task. However, reward functions are directly or indirectly encoded by humans, based on some insight/intuition. Practically, they can be very hard to design, and if poorly designed, can lead to unexpected behvaiors as evident from previous works (Clark \& Amodei (2016), Burda et al. (ICLR 2018)). Our mis-specified reward environment essentially presents a specific case where the intended behavior is for the agent to move towards the goal state, but the environment contains a 'distractive' reward, which can be assumed to be caused due to mis-specification of the reward either due to human error or due to unforeseen properties of the simulation environment. We have clarified this point in the revised draft.
>
> $\textbf{Eq is kind of..}$ The equation number is missing, so we are not entirely sure which equation is being referred to. We dropped the dependence on $\pi$ for simplicity of notations.

---

### Author Response · Authors · 2021-11-22
**General response**

We thank the reviewers for their valuable comments. We have accordingly tried to improve the clarity of several points, which we believe caused confusion regarding the contributions of the work. The main changes are:

(a) We have explicitly mentioned the biased nature of learning the trace function when using neural networks.

(b) Added performance curves for using the trace function as an exploration bonus in the simple environment.

(c) We have tried to clarify the proposition, and moved it to the appendix, where it now stands alone.

(d) Fig 3 was corrected

---

### Decision · Program_Chairs · 2022-01-20

**Decision:**

Reject

**Comment:**

The authors present a method for creating a curriculum for goal-conditioned reinforcement learning. In particular, they propose to use reachability traces to define a sequence of sub-goals that aid learning. During the review process, the reviewers mentioned the novelty of the proposed approach and the intuitive explanations provided by the authors. However, the reviewers also pointed out that the experiments could be more thorough, errors in the theoretical justification of the method as well as simplicity of the evaluation environments, among others. Some of the reviewers increased their score after the authors' rebuttal but it was not enough to advocate for acceptance of the paper. I encourage the authors to incorporate reviewers' feedback in the next version of the paper.